# The Use of a Three-Fluid Atomising Nozzle in the Production of Spray-Dried Theophylline/Salbutamol Sulphate Powders Intended for Pulmonary Delivery

**DOI:** 10.3390/pharmaceutics12111116

**Published:** 2020-11-20

**Authors:** Stefano Focaroli, Guannan Jiang, Peter O’Connell, John V. Fahy, Anne-Marie Healy

**Affiliations:** 1School of Pharmacy and Pharmaceutical Sciences, Panoz Institute, Trinity College Dublin, Dublin 2 D02, Ireland; gjiang@tcd.ie (G.J.); peoconne@tcd.ie (P.O.); healyam@tcd.ie (A.-M.H.); 2Division of Pulmonary and Critical Care Medicine, Department of Medicine and Cardiovascular Research Institute, Health Sciences East, UCSF, 513 Parnassus Avenue, San Francisco, CA 94143, USA; John.Fahy@ucsf.edu; 3SSPC The SFI Research Centre for Pharmaceuticals, School of Pharmacy and Pharmaceutical Sciences, Trinity College Dublin, Dublin 2 D02, Ireland

**Keywords:** spray drying, three-fluid nozzle, dry powder, theophylline, salbutamol sulphate pulmonary drug delivery, dry powder inhaler (DPI)

## Abstract

The aim of this study was to investigate the use of a three-fluid atomising nozzle in a lab-scale spray dryer for the production of dry powders intended for pulmonary delivery. Powders were composed of salbutamol sulphate and theophylline in different weight ratios. The three-fluid nozzle technology enabled powders containing a high theophylline content to be obtained, overcoming the problems associated with its relatively low solubility, by pumping two separate feed solutions (containing the two different active pharmaceutical ingredients (APIs)) into the spray dryer via two separate nozzle channels at different feed rates. The final spray-dried products were characterized in terms of morphology, solid-state properties and aerosolization performance, and were compared with an equivalent formulation prepared using a standard two-fluid atomising nozzle. Results confirmed that most of the powders made using the three-fluid atomising nozzle met the required standards for a dry powder inhaler formulation in terms of physical characteristics; however, aerosolization characteristics require improvement if the powders are to be considered suitable for pulmonary delivery.

## 1. Introduction

Pulmonary delivery of multiple active pharmaceutical ingredients (APIs) has received increasing attention in recent years for the treatment of chronic obstructive pulmonary disease (COPD) [1]. Generally, a combination therapy can be obtained by using different inhalers; however, a combination dosage form delivered via a single inhaler represents a more attractive strategy from a patient compliance point of view, in comparison with the sequential administration of individual drug doses [2,3,4].

COPD treatment by pulmonary drug delivery is usually obtained by inhalation of different classes of drugs, such as β2 agonists, corticosteroids and anticholinergics [5]. Alongside the pulmonary delivery of such therapeutic agents, either intravenous or oral administration of theophylline (TH) is prescribed in 35% of patients affected by COPD in parallel with their inhalation therapy [6,7]. The effectiveness of TH is due to its bronchodilatory and bronchoprotective properties, as well as its anti-inflammatory and immunomodulatory effects. The mechanism of action is based on the non-selective inhibition of phosphodiesterase and a subsequent increase in intracellular cAMP, which relaxes the smooth muscle of the airways [8,9]. However, the low selectivity of TH towards different phosphodiesterase isoforms is the basis for its narrow therapeutic index, and its systemic effect often causes a wide range of adverse reactions such as nausea, vomiting, abdominal pain, metabolic acidosis, seizures, and arrhythmias [7,10]. Nevertheless, TH is still in receipt of a great deal of attention in the scientific community since it presents undeniable advantages, including its low cost compared to other long-term maintenance medications and the ability to control chronic asthmatic and COPD diseases [7]. There is potential to reformulate this API as an inhaled medicine with the aim of reducing its side effects. This approach should open up benefits that would not be possible using the more common routes of delivery (i.e., oral or intravenous). Other authors have previously formulated TH for pulmonary delivery. For example, Raeburn and Woodman in 1994 intratracheally applied TH as a dry powder to the airways of anaesthetised guinea pigs, showing the effectiveness of the drug in terms of smooth muscle relaxation and anti-inflammatory properties at a concentration that would be predicted to have no systemic toxicity [11,12]. More recently Zhu and co-workers formulated TH as a pressurised metered dose inhaler (pMDI) solution. Their study showed a good TH in vitro lung deposition profile from the pMDI, but the main disadvantage of this delivery platform was that the dosing was limited to a few micrograms [13,14].

Among the inhaled dosage forms for pulmonary therapy, dry powder-based medications seem to be the most promising and attractive, because of their improved physicochemical stability in comparison with liquid-based formulations [15,16].

Furthermore, dry powder inhaler (DPI) devices present more advantages than the most commonly used pMDIs, such as the possibility to obtain high administered doses, absence of propellants and breath-actuated operation [17,18,19]. A few studies have been previously published that are concerned with engineering inhalable theophylline particles, alone or in combination with other drugs [10,11,14,20,21,22], but the approaches described to obtain respirable particles are characterized either by a number of sequential steps being involved in the formulation and manufacturing process, or by the use of relatively large amount of excipients. In the first case the cost of the whole process might rise considerably in terms of time and reagent utilisation (especially on a large scale), whereas the use of large amounts of inactive ingredients in the formulation may hinder effective delivery of the drug, since dose limitation problems could occur [23,24].

In this paper, we aimed to formulate and manufacture an excipient-free combination of salbutamol sulphate (SS) and theophylline in the form of an inhalable dry powder intended for the treatment of COPD.

Salbutamol sulphate was chosen since it is a widely used short-acting β2-agonist in the symptomatic relief of asthma and COPD [25] and is frequently prescribed as maintenance therapy. A combination of inhaled SS and TH could provide greater efficacy for patients who remain symptomatic on a monotherapy with SS [26]. The effectiveness of this combination therapy has been already established in the literature; for example, Nishimura et al. demonstrated a superior bronchodilation effect by combining inhaled SS and orally delivered TH compared to the use of the individual agents alone [27].

Powders were prepared by spray drying, a one-step process that allows the conversion of a liquid feed into a dried particulate [28,29]. Spray drying is also a cost-effective method and offers the possibility of engineering particles with customized properties [30,31]. As previously stated, the first step of this process is the conversion of the feed into micronized droplets by atomisation. This atomisation process is commonly obtained by pumping a solution, or a suspension, containing one or more APIs, through a pneumatic device in which a compressed atomising gas flows separately; the two phases meet one another at the end of a nozzle and the liquid feed stream is broken into droplets upon contact with the gas. This atomisation device is also known as “pneumatic nozzle” or, more commonly, a “two-fluid nozzle” (2-F) [28,32]. An alternative system is represented by the use of the three-fluid nozzle (3-F), in which two different liquid feeds are pumped through two separate liquid passages or channels where they are atomized by the gas provided by a third channel [33]. On foot of its design, the application of the three-fluid nozzle may offer some advantages over the two-fluid nozzle for the production of a dry powder combination product: Firstly, it is possible to dissolve two APIs in two different solvents prior to spray drying, overcoming problems associated with differing API solubilities; and, secondly, it allows the use of high concentration solutions with a consequent reduction in solvent waste and higher process yields.

A particular challenge associated with theophylline is its low solubility in commonly used solvents such us water, acetone and ethanol [34], and thus it may be difficult to obtain a composite dry powder containing a sufficient amount of TH in the final product, even if a 3-F nozzle is used. However, in the current study we proposed and tested an alternative approach to overcome this limitation, by taking advantage of the possibility of using different feed flow rates for the inner and outer channels of the 3-F nozzle. Thus, the SS-containing feed solution was fed into the inner channel of the nozzle and the TH-containing feed solution was pumped into the outer channel of the nozzle, with different feed rates providing different feed rate ratios. The formulations obtained using this approach were then characterized in terms of morphology, solid-state properties and aerosolization performance, and comparisons were made with formulations prepared using a standard two-fluid atomisation nozzle.

## 2. Materials and Methods

### 2.1. Materials

Salbutamol sulphate (SS) was purchased from Kemprotec Ltd. (Cumbria, UK). Theophylline (TH, anhydrous, ≥99%) was purchased from Sigma-Aldrich, Dublin, Ireland. Ultra-pure water was prepared by a Millipore Elix advantage water purification system. Ethanol was provided by Lennox Laboratories (Dublin, Ireland). Chloroform and acetone were purchased from Sigma-Aldrich (Dublin Ireland).

### 2.2. Preparation of Salbutamol Sulphate/Theophylline Powders by Spray Drying

Formulations containing SS and TH were prepared using a Büchi B-290 spray dryer (Büchi, Flawil, Switzerland). Preliminary experiments were conducted to enable particles in the respirable size range to be obtained, by altering spray dryer inlet temperature, spray dryer aspiration pump rate and feed flow rates. In subsequent experiments, for all results reported, an inlet temperature of 140 °C and an aspiration rate of 90% were used. Other applied process conditions in terms of feed flow rate and spray dryer configuration (open mode or close mode) are listed in Table 1. A B-295 inert loop and a dehumidifier (Büchi, Flawil, Switzerland) were used when ethanol, chloroform and acetone were included in the feed solution. When the three-fluid nozzle (2 mm nozzle tip diameter, 2.8 mm nozzle cap diameter) was used, TH and SS solutions were fed separately into the outer and inner channels of the nozzle respectively. SS was dissolved in pure water at three different concentrations namely 2.25%, 2%, 1.75% (*w*/*v*) whereas TH was dissolved in water, ethanol, chloroform or acetone at a concentration of 0.25% in all the experiments. The final TH/SS weight ratio of the dry powders obtained differed depending on the SS feed concentration and the flow rate ratio between the TH and SS solutions. In this work all the samples were obtained by spray drying 80 mL of TH solution at a constant feeding rate (i.e., 3.40 mL/min), whereas 80, 160 and 240 mL of SS solution were pumped at the flow rates of 3.40, 1.70 and 1.13 mL/min in order to achieve a TH/SS flow rate ratio of 1:1, 2:1 and 3:1 respectively.

The above approach enabled various drug composition ratios to be obtained in the final product, as listed in Table 1. Since the Büchi spray dryer is equipped with only one peristaltic pump, an external pump (Ismatec, Glattbrugg-Zurich, Switzerland) was utilized in order to pump the TH solution through the nozzle outer channel. Formulations produced using a standard two-fluid nozzle (0.7 mm nozzle tip diameter, 1.5 mm nozzle cap diameter) were obtained by dissolving both TH and SS in water, keeping the TH concentration constant at 0.25% *w/v* and by varying the SS amount in the solution in order to obtain a SS/TH composition ratio of 7:3, 8:2 or 9:1 (% *w*/*w*). The solutions were pumped into the nozzle at 3.4 mL/min. Salbutamol sulphate and theophylline were also spray-dried alone using a standard two-fluid nozzle in an open mode configuration. Both drugs were dissolved in water at a concentration of 2.25 and 0.25% (*w*/*v*) respectively. The solutions were fed into the spray dryer at 3.40 mL/min keeping the inlet temperature at 140 °C, the gas feed rate at 50% (corresponding to 600 L/h) and the aspiration rate at 90%. A schematic representation of the two-fluid and three-fluid nozzles is depicted in Figure 1. Spray-dried SS and TH physical mixes in weight ratios of 9:1, 8:2 and 7:3 (*w*/*w*) were prepared by gently mixing about 200 mg of powder blends in a mortar using a spatula for 2 min.

### 2.3. Scanning Electron Microscopy (SEM)

SEM micrographs of spray-dried materials were acquired using a Zeiss Supra Variable Pressure Field Emission Scanning Electron Microscope (Oberkochen, Germany) equipped with a secondary electron detector. The samples were fixed on aluminium stubs using double-sided adhesive tape and sputter-coated with gold. Visualization was performed at 5 kV and photomicrographs were taken at different magnifications in more than one region of the sample.

### 2.4. Yield

Spray drying yields were calculated based on the weight of powder collected, expressed as a percentage of the weight introduced into the feed solution, giving the yield percent by weight (% *w*/*w*).

### 2.5. Particle Size Measurement

Particle size distributions were obtained using a Mastersizer 3000 laser diffraction instrument (Malvern Instruments, Malvern, UK) with a dry powder dispersion accessory (Malvern Aero S) and approximately 10 mg of powder. In order to achieve an obscuration between 0.5–6%, the dispersive air pressure was 3 bar, and vibration feed rate was 75%. The data was evaluated using Mastersizer 3000 software (Malvern Instruments, Malvern, UK, version 3.62). The particle size was represented by the D50, D90 and D10 parameters. The D50 is the median particle size of the volume distribution, while the D90 and D10 are the particle sizes corresponding to 90% and 10% of the cumulative percentage undersize volume distribution, respectively. The range of the particle size distribution was expressed by the span value, shown by Equation (1) below
Span = (D90 − D10)/D50(1)
Each result reported is the average of three determinations.

### 2.6. Fourier Transform Infrared Spectroscopy (FTIR)

IR spectra were obtained using a PerkinElmer Spectrum one FT-IR Spectrometer (PerkinElmer, Dublin, Ireland) and evaluated by Spectrum v5.0.1 software (PerkinElmer, Dublin, Ireland). All spectra were baseline corrected, and an average of 10 scans with a resolution of 4 cm^−1^ over a wavenumber range of 4000–650 cm^−1^ was used for all dried powders.

### 2.7. Powder X-ray Diffraction (PXRD)

PXRD analysis was performed using a Rigaku Miniflex II desktop X-Ray diffractometer (Rigaku, Tokyo, Japan) with Ni-filtered Cu Kα radiation (1.54 Å). Powders were placed on a low background silicon sample holder and the PXRD patterns were recorded from 5° to 40°on the 2-theta scale at a step of 0.05°/s. The X-ray tube, composed of Cu anode (λCuKα = 1.54 Å), was operated under a voltage of 30 kV and current of 15 mA. PXRD patterns were recorded at least in duplicate.

### 2.8. Thermal Analysis: Thermogravimetric Analysis and Modulated Differential Scanning Calorimetry (MDSC)

Thermogravimetric analysis (TGA) was performed using a TGA Q50 (TA Instruments, Dublin, Ireland) to determine the residual solvent content (RSC) contained in samples after spray drying. Samples were placed into open aluminium pans; a temperature range of 25 to 200 °C was employed with a heating rate of 10 °C/min under nitrogen purge and the RSC was defined as the weight loss between 25 and 150 °C. The values presented are the average of three analyses. Modulated differential scanning calorimetry measurements were carried out under nitrogen purge using a MDSC Q200 instrument (TA Instruments, Dublin, Ireland) calibrated with indium. Powder samples were placed in aluminium pans, sealed and heated over the temperature range of 0 to 170 °C. The modulation parameters chosen to ensure separation of reversing and non-reversing events were ±0.8 °C modulation amplitude and a modulation period of 60 s, with a heating ramp rate of 5 °C/min. The values reported represent an average of three analyses. Both TGA and MDSC systems were controlled by Q Advantage software (version 5.5, TA Instruments, Dublin, Ireland).

### 2.9. In Vitro Aerosol Deposition Studies Using the Next Generation Impactor (NGI)

The pulmonary deposition of the spray-dried powders was estimated in vitro using a Next Generation Impactor (NGI, Copley Scientific Limited, Nottingham, UK). The flow rate was adjusted to achieve a pressure drop of 4 kPa in the dry powder inhaler (Low resistance RS01, Plastiape, Osnago, Italy) and the time of aspiration was adjusted to obtain 4 L of air flow. The NGI cups were lined with 47 mm diameter GE Healthcare Whatman filter papers (Thermo Fisher Scientific, Dublin, Ireland) wetted with 1 mL of deionized water to minimize particle bounce. The deposited powders were dissolved in water and quantified using HPLC (as detailed in Section 2.10). The dry powder inhaler was loaded with a no. 3 HPMC capsule loaded with 10 mg of powder for each formulation tested. Deposition profiles for each formulation were determined in triplicate and the results presented are the average results of the replicated analyses. The total amount of particles with aerodynamic diameters smaller than 5 and 3 µm was calculated by interpolation, from the inverse of the standard normal cumulative mass distribution less than stated size cut-off against the natural logarithm of the cut-off diameter of the respective stages [35]. These amounts were considered the fine particle fraction (FPF) below 5 µm and 3 µm and are expressed as a percentage of the emitted recovered dose (ERD). The mass median aerodynamic diameter (MMAD) of the powder was determined from the same plot and is the particle size corresponding to the 50% point of the cumulative distribution. The geometric standard deviation (GSD) was calculated from Equation (2):
D = √(SizeX/SizeY)(2)
where X is the particle size corresponding to the 84% point and size Y is the particle size corresponding to the 16% point of the cumulative distribution [36,37].

### 2.10. High-Performance Liquid Chromatography (HPLC)

The HPLC method for the quantification of SS and TH was adapted from the British Pharmacopeia [38]. A Breeze™ HPLC system (Waters, Milford, CT, USA) equipped with Waters 1525 binary pump with an in-built degasser, Waters 717 plus autosampler and Waters 2487 dual λ absorbance detector was used. Isocratic chromatographic separation was carried out at room temperature using a 125 × 4.6 mm (5 µm particle size) Inertsil ODS3 column (GL Sciences, Eindhoven, The Netherlands). The mobile phase comprised of 22% *v*/*v* acetonitrile and 78% *v*/*v* of an aqueous solution containing 0.264% *w*/*v* of 1-heptanesulfonic acid sodium salt and 0.25% *w*/*v* potassium phosphate monobasic adjusted to pH 3.7. The flow rate and injection volume used were 1.0 mL/min and 20 µL, respectively. UV detection was performed at 220 nm. The retention times for TH and SS were 2.5 and 4.0 min, respectively. The limit of detection (LOD) and quantification (LOQ) were obtained as the concentration with a signal-to-noise ratio at 3 and 5, respectively. LODs were found to be 12 and 5 µg/mL for SS and TH, respectively, whereas SS LOQ was 20 µg/mL and TH LOQ was 8 µg/mL.

### 2.11. Statistical Analysis

The two-sample *t*-test (comparison of two data sets) and one-way ANOVA followed by posthoc Tukey’s test (comparison ≥ 3 sets of data) were used to test the statistical significance of the samples when comparisons were made. Differences were found to be statistically significant when a confidence level of 95% (i.e., *p*-value ≤ 0.05) was obtained. OriginPro software (version 9, OriginLab, Northampton, MA, USA) was used to perform the statistical analyses.

## 3. Results and Discussion

### 3.1. General Considerations and SEM Analysis

The use of a three-fluid nozzle allows simultaneous spraying of two different feed solutions into the drying chamber and the consequent production of a particulate comprising the combination of solutes solubilized in the feeds. The final powder composition obviously depends on the amount of solutes and generally reflects the concentration ratio of the two solutions. Nevertheless, the formulation of compounds that have low solubilities in both hydrophobic and hydrophilic solvents represents a challenge, since it may be difficult to obtain a binary system containing an adequate amount of poorly soluble drug in the final product, especially when a high delivered dose is required. In this study, salbutamol sulphate was solubilized in water, since it represents the most suitable solvent for the compound and is also suitable for spray drying. In contrast, theophylline has limited solubility in water in its unionized form (7360 mg/L) [31]. In order to increase TH solubility, it was initially solubilized in a 0.1 M NaOH solution prior to spray drying, but the final product resulted in a sticky powder which was not useful for practical purposes (data not shown). For this reason, a number of organic solvents were tested, including ethanol, acetone and chloroform, as a suitable solvent for TH. However, even with these solvents, TH was still hard to solubilize, and no significant improvement in TH solubility was obtained. Considering these preliminary results, all the experiments in this study were performed using a TH concentration of 0.25% *w*/*v* in all solvents, with the aim of comparing the effect of such different solvents on the formulation properties. On the other hand, the SS concentration was varied (in water) according to the drug composition required in the final product. It is worth noting that the solvents chosen in this study allowed us to monitor the effect of liquids having different affinities with water, since, as well as using water alone, both water miscible (ethanol and acetone) and water immiscible (chloroform) solvents were used for the TH feed solution.

With respect to the SS/TH 9:1 (*w*/*w*) formulations (3F1, 3F4, 3F7, 3F10; Table 1), the powders were obtained by pumping the same volume of feed solutions having a concentration of 2.25% and 0.25% *w*/*v* of SS and TH, respectively. In this case, both the feeds were pumped at the same rate through the nozzle channels, so that the relative amount of drug in the final products should reflect the actual ratio between the feed solution concentrations (i.e., 2.25/0.25 or 9/1). Nevertheless, this approach has some limitations when used for the production of powders containing a higher amount of theophylline. In fact, as indicated above, the TH solubilisation limit did not allow a further increase of drug concentration in the feed solution and the only way to increase the amount of TH in the final product was to reduce the SS concentration. To overcome this drawback, an alternative strategy was proposed: The formulations containing 20% and 30% (*w*/*w*) of TH produced by the 3F-nozzle were obtained by keeping the feed flow rate of the TH solutions constant, while decreasing the feed flow rates of the SS to one half and one third of the TH solution feed rate, in order to obtain a TH/SS feed flow rate ratio of 2:1 and 3:1, respectively. Such an approach enabled the amount of TH within the dry products to be increased, while maintaining the SS concentrations relatively high (2% and 1.75% *w*/*w* respectively).

In this study, as well as using the three-fluid nozzle approach to adjust the ratio of components in a composite powder, the starting hypothesis was that the three-fluid nozzle could be used to formulate both APIs into a core-shell particulate [33]. The intention was to embed an amorphous SS core (pumped in the inner channel of the three-fluid nozzle) within a crystalline TH shell that would be able to protect the core from environmental moisture, in order to provide for better physical stability of the amorphous material, as well as potentially improve the flow characteristics of the powder [23,39].

SS spray dried alone from water, using the 2F nozzle, resulted in spherical particles with dimpled surfaces (Figure 2B), while TH spray dried alone from water resulted in aggregates of nano-crystals (Figure 2A). SEM analysis of the powders obtained using the three-fluid nozzle (Figure 2) revealed that, while in some instances a uniform particle morphology was achieved (Figure 2C,D,L), similar to the morphology of SS spray-dried alone, in other cases powders appeared to be composed of a mixed binary system comprised of TH nano-crystals and separate SS dimpled particles. Generally, the TH crystals were adsorbed onto the particle surfaces (Figure 2E–H), but they also formed relatively large stand-alone clusters in some cases (Figure 2I–K). This latter morphology seemed to be solvent dependent, with such structures being mainly found when chloroform was used to solubilise TH (Figure 2I–K). This may be explained by the fact that the low miscibility of chloroform with water does not allow for intimate contact between SS and TH during the spray drying process.

Figure 2O–Q shows the formulations produced by the two-fluid nozzle in which both drugs were solubilised together in water in one common feed solution. It is evident that the apparent propensity of the powders to deliquesce (as shown by loss of discrete particle morphology and the merging of particles into larger solid masses (Figure 2P,Q)) is directly proportional to the concentration of TH in the formulations. As stated previously, the use of the two-fluid nozzle necessitated a reduction in the total solid content in the feed solution, in order to maintain the correct proportions of APIs in the final product. The reduction in total solid content influences the drying efficiency of the droplets resulting in relatively “wet” products, as shown by the relatively high residual solvent content of sample 2F-3 in Table 2 (4.7% *w*/*w*), whereby non-evaporated bulk water may dissolve soluble compounds such as SS [40]. Another reason for the observed deliquescence and morphology might be the high tendency for those formulations to absorb water from the environment. In general, the amount of water absorbed is proportional to the volume/weight of the amorphous content in the solid [41].

### 3.2. PXRD and FTIR Analysis

The PXRD diffractograms of the formulations processed with the two-fluid nozzle, 2F_1-3, showed diffuse halo patterns, characteristic of fully XRD amorphous materials (Figure 3). However, all the products obtained using the three-fluid-nozzle were characterized by a certain degree of crystallinity, as demonstrated by the presence of a peak at 12° (2 Theta), which corresponds to the main peak of crystalline TH (Appendix A); the intensity of the peak being proportional to the amount of TH in the final product. It is possible that such partial crystallinity makes the powders less susceptible to moisture uptake. The absence of this peak in the formulations obtained using the two-fluid nozzle is probably due to the propensity of both APIs to form a solid glass solution when they are solubilized in and spray-dried from the same feed solution. On the other hand, the flowing of two different feed solutions in separated channels (in the three-fluid nozzle) precluded full contact between the two APIs to some extent, and, consequently, a binary, phase separated system was obtained in which TH and SS precipitated/solidified separately during the drying process. Furthermore, the miscibility of the solvents seemed to play a role in determining the solid state characteristics of the spray-dried material. In fact, the intensity of the TH-associated Bragg peak is higher when TH was solubilised in chloroform (samples 3F_7–9). This result matched with the findings obtained by SEM detailed above regarding the amount of TH crystals observed: the use of non-miscible solvents (i.e., chloroform and water) hindered intimate contact between the molecules of the APIs, resulting in a powder containing more TH crystals in comparison with those formulations produced from solutions of TH in water-miscible solvents (i.e., ethanol and acetone), in which there appears to be a greater propensity for TH to interact with SS producing a glassy matrix.

Appendix A show the FTIR spectra of raw materials and spray dried samples. The conversion of SS (unprocessed) to the amorphous form (spray dried) resulted in peak broadening. The physical mixtures and all the spray dried samples presented the same bands as the raw materials, indicating that any interaction between theophylline and salbutamol sulphate is limited, even for the sample processed using the two-fluid nozzle, where XRPD data indicate the probable formation of a solid solution of the two APIs.

### 3.3. Product Yield

Results of powder physicochemical characteristics and spray dryer outputs are listed in Table 2. The yields of the samples prepared using the three-fluid nozzle were in the range 13–90%; the substantial losses being mainly caused by the high deposition of droplets and particles on the drying chamber and cyclone walls, especially when the formulations were spray-dried in a TH/SS ratio of 1:9 (except when both drugs were solubilised in water, where the product yield was relatively high in all cases).

### 3.4. Particle Size Distribution

Most spray-dried powders showed a monomodal particle size distribution, with a D50 ranging between 3.1 and 5.9 µm, which indicates that most of them are suitable for pulmonary drug delivery [23]. However, some samples displayed high D90 values, probably due to the formation of particle aggregates, as indicated by the SEM images, which might result in variations in powder aerosol behaviour and performance [20]. This aggregation tendency can be attributed to cohesion between the particles, as is inherent to spray-dried fine particulate drugs. No obvious correlations were found between this output (D50) and the spray drying parameters used.

### 3.5. Thermal Analyses: mDSC and TGA

DSC thermograms are shown in Figure 4A–C, Appendix A. The upper temperature of the scans was 170 °C which represented the degradation onset of the samples. The total heat flow thermograms (Appendix A) showed a broad endothermic event due to the removal of residual solvent in all samples (except for the spray-dried TH alone). The enthalpy associated with this event was particularly high when both drugs were solubilised in water. This result was not surprising since the evaporation enthalpy of water (2256 kJ/kg) is much higher than the other solvents used (846, 518 and 247 kJ/kg for ethanol, acetone and chloroform, respectively) [42], and more energy is required to drive off the excess liquid.

In the reverse heat flow signal (Figure 4), a glass transition was observed in the temperature range of 100 and 117 °C for all spray-dried samples obtained by the 3F nozzle. The glass transition temperature (T_g_) of SS spray-dried alone was 108 °C, whereas spray-dried TH did not show any event in the temperature range considered. Thus, it is reasonable to argue that the T_g_ values for the various spray-dried samples were solely ascribed to the SS glass transition. The formulations produced using the standard two-fluid nozzle revealed a decrease in the powder T_g_s as the SS content, and correspondingly RSC, increased (Table 2), reaching 93 °C in the sample 2F_3.

Residual moisture or solvent content in the range of 2.1% *w*/*w* to 3.6% *w*/*w* were determined by TGA. In the samples containing a TH/SS ratio of 3:7, the *t*-test revealed a statistical difference between the particles made using the three-fluid nozzle (p = 0.0014; 0.0119; 0.0011 and 0.0137 for ST3, ST6, ST9 and ST12 respectively) and that obtained using the two-fluid nozzle (2F_3). The higher solvent amount in 2F_3 might be due to the low total solid content in the feed solution and the consequent reduction of drying efficiency of the droplets related to the higher amount of water to be evaporated [43]. This finding also explains the low T_g_ value determined for sample 2F_3 as a result of the plasticizing effect of water [44].

Nevertheless, the T_g_ values of all the sample tested are expected to promote the physical stability of amorphous powders during storage, as long as the T_g_ of the formulation is approximately 50° K greater than the storage temperature [45,46].

### 3.6. In-Vitro Deposition Profile

The next generation impactor (NGI) was used to assess the aerodynamic performance of the spray-dried powders using a low resistance RS01 Plastiape dry powder inhaler device. For this experiment only, the formulations made of a TH/SS ratio of 7:3 (*w*/*w*) were examined: Namely samples 3F_3, 3F_6, 3F_9, 3F_12 and 2F_3. The mass deposition profiles of SS and TH of the above-mentioned formulations, as well as their key aerosolization parameters, are shown in Figure 5 and Table 3 and Table 4. By studying the aerosolization profiles, it is clear that formulation 2F_3 presented the most suitable values for achieving good pulmonary delivery. In fact, the MMAD of both drugs was within the range 1–5 µm, which represents the desirable range of particle size to obtain an appropriate deposition in the lung. Furthermore, formulation 2F_3 showed higher FPF values than other powders. In this case, the similar deposition profile of the SS and TH suggested that the two APIs will most likely deposit in the same regions of the lungs.

The co-deposition of both drugs was probably due to their pre-mixing in the same feed solution before being subjected to spray drying and the consequent formation of a solid solution as indicated by the XRPD results. On the other hand, for the formulations prepared using the three-fluid nozzle there was a significant reduction in the emitted FPFs (relative to that prepared using the two-fluid nozzle) as well as higher MMAD and GSD values. In addition, the value of those parameters differed considerably for the two APIs contained in the same formulation. A possible explanation for these differences is that the SS and TH were fed from two separate liquid passages and met at the tip of the three-fluid nozzle. Hence, the mixing time may be insufficient to generate a homogeneous mixture of the two APIs, ultimately resulting in non-identical in-vitro deposition of salbutamol sulphate and theophylline. Interestingly the MMADs and FPFs for TH were lower and higher, respectively, than for SS, indicating that the SS particles might act as a carrier for the delivery of theophylline crystals. Among all the formulations prepared using the three-fluid nozzle, sample 3F_3 (where TH was solubilised in ethanol) seemed to be the most promising in terms of aerosolization properties. Nevertheless, those results were obtained assuming that the particle compositions reflected the theoretical values of their calculated weight ratio, namely 7:3. In the light of this, further investigations and powder optimisation are required.

## 4. Conclusions

Combination dosage forms constituted of theophylline and salbutamol sulphate intended for pulmonary delivery were produced by spray drying using a three-fluid nozzle. This approach enabled the manufacture of powders containing different drug ratios in the final product, overcoming the issues related to theophylline’s low solubility in commonly used solvents, such as water, ethanol, chloroform and acetone. Results indicated that the powder particle morphology was solvent and TH/SS ratio dependent. In addition, the products were composed of relatively homogeneous spherical particles or, in some cases, of a binary system in which the theophylline crystals were either adsorbed onto the surfaces of the salbutamol sulphate particles or assembled together in clusters. The physicochemical characteristics (T_g_, particle size distribution) of most of the spray-dried powders suggested their suitability as DPI formulations; however, in vitro deposition profiles of the powders obtained by the three-fluid nozzle were still not sufficient to achieve a proper lung delivery, and further optimisation studies will be required to improve their aerodynamic properties.

## Figures and Tables

**Figure 1 pharmaceutics-12-01116-f001:**
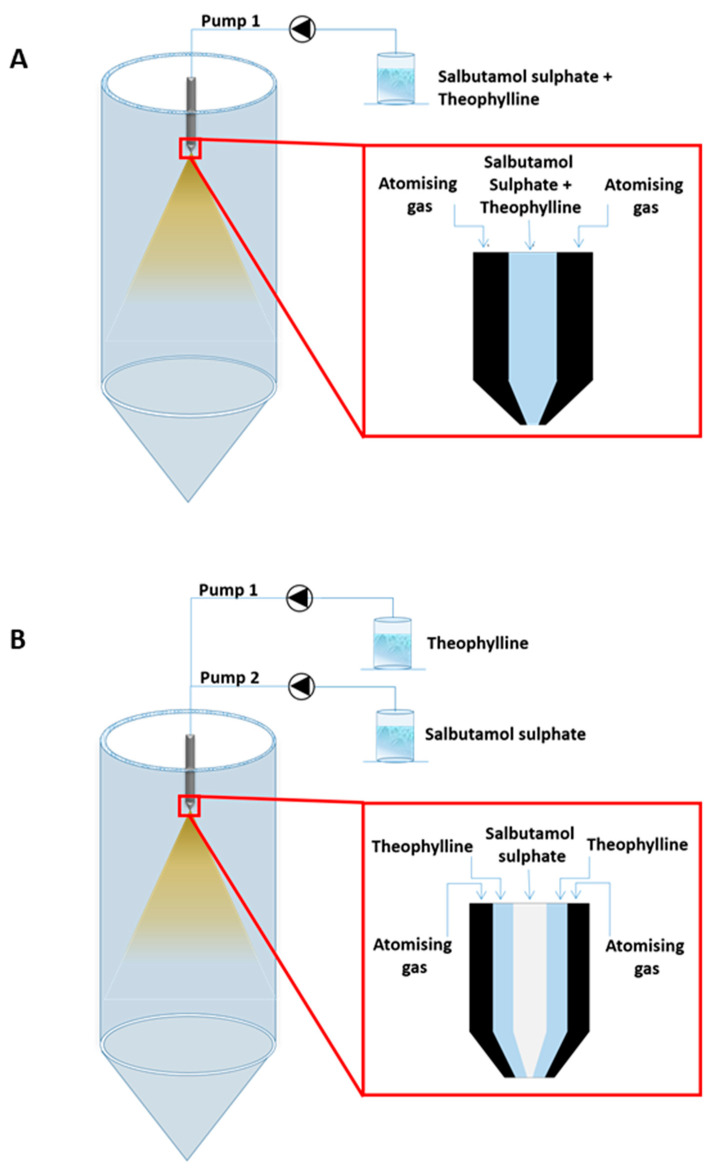
Schematic representation of (**A**) the two-fluid nozzle and (**B**) the three-fluid nozzle.

**Figure 2 pharmaceutics-12-01116-f002:**
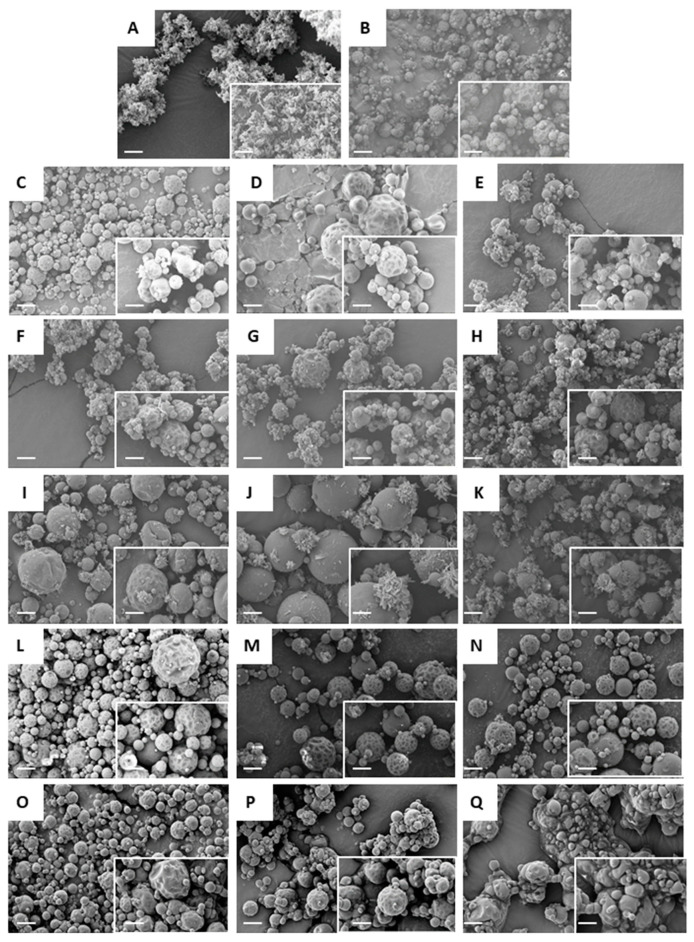
SEM images of the spray-dried formulations. (**A**) Spray-dried theophylline from water; (**B**) spray-dried salbutamol sulphate from water; (**C**–**N**) samples prepared using the three-fluid nozzle 3F_1-12 (as per Table 1); (**O**–**Q**) samples prepared using the two-fluid nozzle 2F_1-3 (as per Table 1). Bars correspond to 4 µm. Insert bars correspond to 2 µm.

**Figure 3 pharmaceutics-12-01116-f003:**
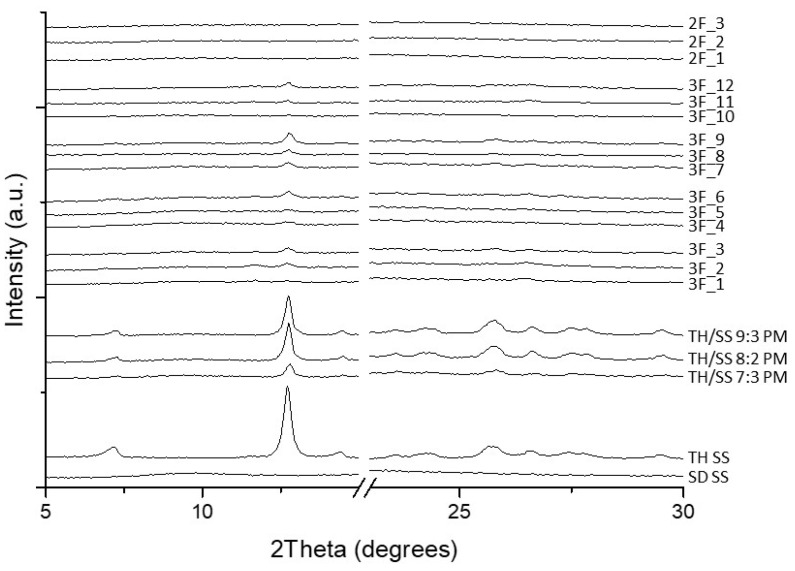
Powder X-ray Diffraction (PXRD) patterns of spray-dried samples and physical mixtures.

**Figure 4 pharmaceutics-12-01116-f004:**
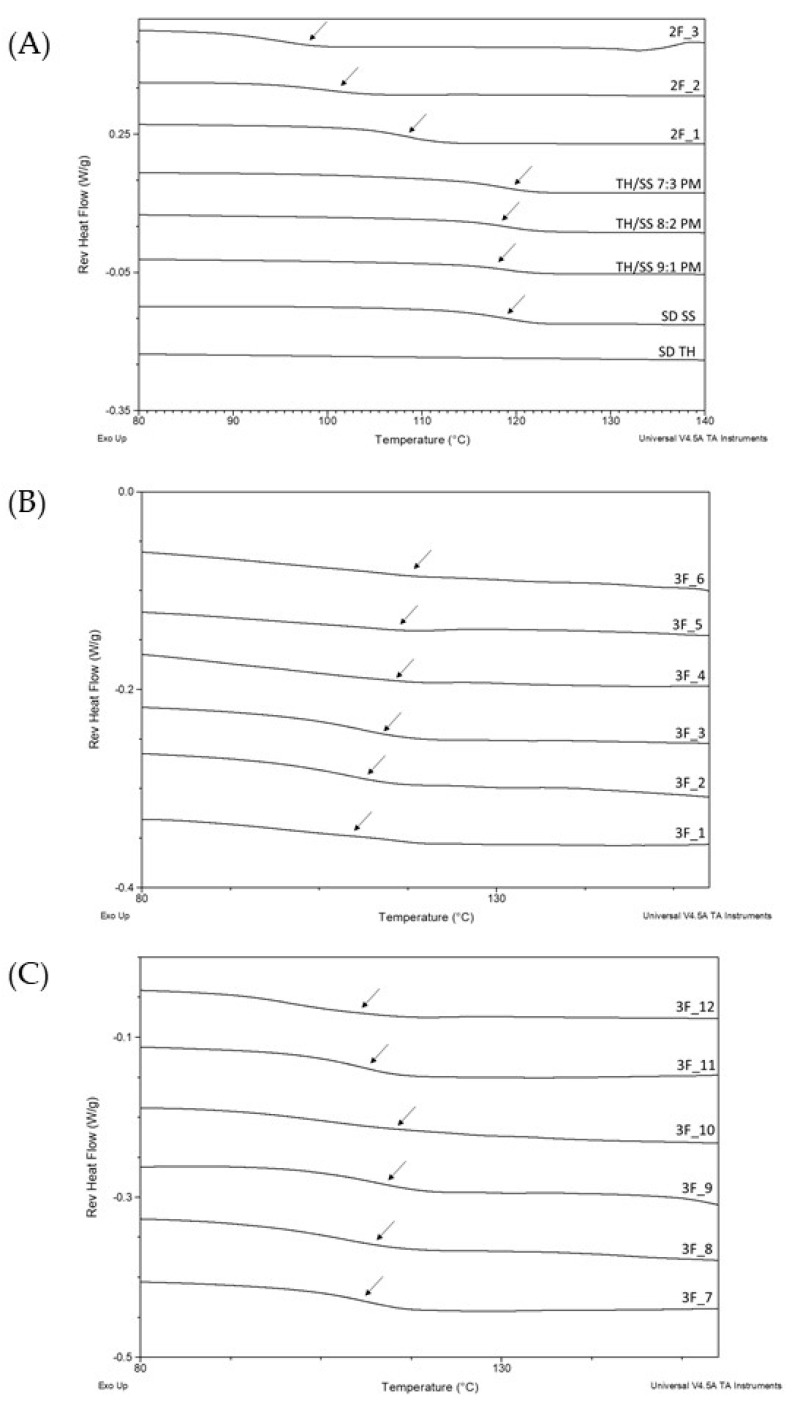
Representative reversible heat flow Modulated Differential Scanning Calorimetry (MDSC) thermograms of powders, including (**A**) spray-dried theophylline; spray-dried salbutamol sulphate, spray-dried SS and TH physical mixture, and samples 2F_1–3; (**B**) samples 3F_1–6; (**C**) samples 3F_7–12. Arrows indicate the mid-point of the T_g_ value.

**Figure 5 pharmaceutics-12-01116-f005:**
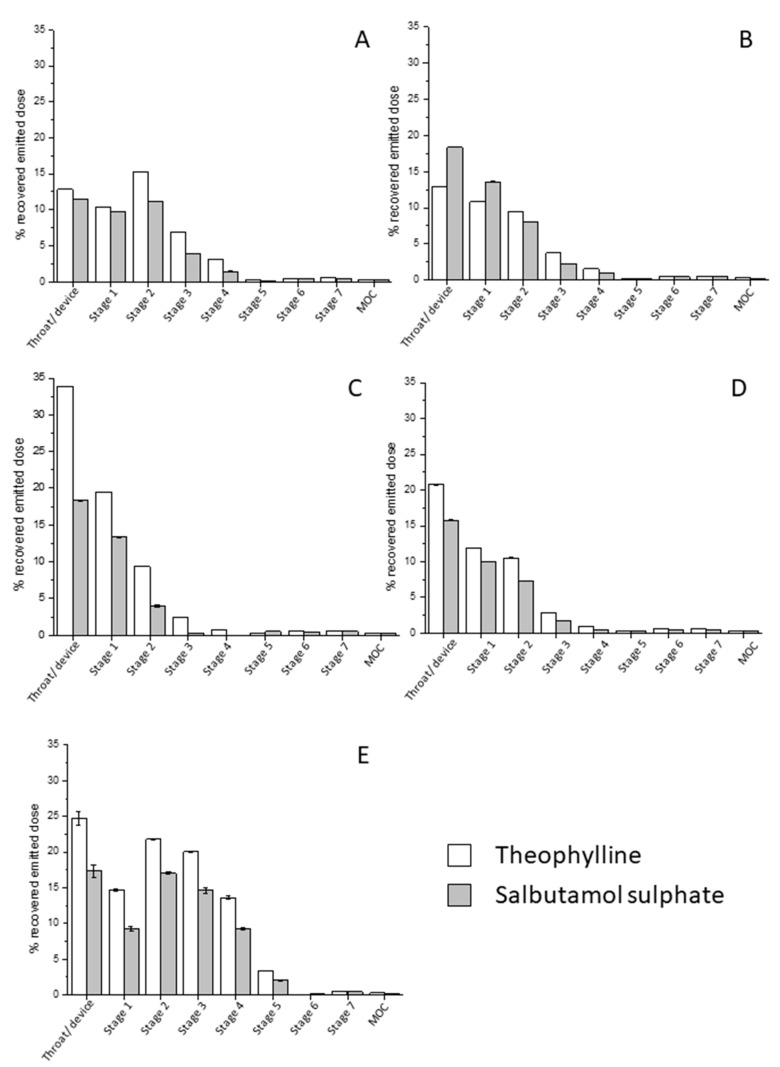
Next Generation Impactor (NGI) mass deposition profiles of spray-dried formulations, (**A**): 3F_3; (**B**): 3F_6; (**C**): 3F_9; (**D**): 3F_12; (**E**): 2F_3.

**Table 1 pharmaceutics-12-01116-t001:** Sample IDs (3F = three-fluid nozzle was used; 2F = two-fluid nozzle was used) and the variable parameters used to obtain the formulations.

Sample ID	Theophylline Solvent	TH/SS Ratio (*w*/*w*)	Atomizing Gas Flow Rate (%)	Salbutamol Sulphate Solution Feed Flow Rate (mL/min)	TH/SS Flow Rate Ratio	Spray Dryer Configuration
3F_1	Ethanol	1:9	60	3.4	1	Closed
3F_2	Ethanol	2:8	60	1.7	2	Closed
3F_3	Ethanol	3:7	60	1.1	3	Closed
3F_4	Acetone	1:9	60	3.4	1	Closed
3F_5	Acetone	2:8	60	1.7	2	Closed
3F_6	Acetone	3:7	60	1.1	3	Closed
3F_7	Chloroform	1:9	60	3.4	1	Closed
3F_8	Chloroform	2:8	60	1.7	2	Closed
3F_9	Chloroform	3:7	60	1.1	3	Closed
3F_10	Water	1:9	60	3.4	1	Open
3F_11	Water	2:8	60	1.7	2	Open
3F_12	Water	3:7	60	1.1	3	Open
2F_1	Water ^a^	1:9	60	3.4 ^b^	NA	Open
2F_2	Water ^a^	2:8	50	3.4 ^b^	NA	Open
2F_3	Water ^a^	3:7	40	3.4 ^b^	NA	Open

^a^ Both drugs were solubilized in the same feed solution. ^b^ The value represents the feed flow rate of the single solution spray-dried using a standard two-fluid nozzle. TH: Theophylline. SS: Salbutamol sulphate.

**Table 2 pharmaceutics-12-01116-t002:** Main physical characteristics of spray-dried samples.

Sample ID	T_out_ (°C)	Yield (%)	Particle Size (µm)		Span	Residual Solvent Content (Weight %)	T_g_ (°C)
			D_50_	D_90_			
3F_1	76	55.7	3.7 ± 0.1	9.7 ± 0.4	2.3	3.6 ± 0.1	114 ± 1
3F_2	75	84.5	3.1 ± 0.1	7.5 ± 0.9	2.2	3.5 ± 0.1	108 ± 5
3F_3	76	78.1	3.2 ± 0.1	7.1 ± 0.1	1.9	2.5 ± 0.1	106 ± 5
3F_4	67	12.8	4.5 ± 0.2	102.7 ± 31.6	22.8	2.4 ± 0.1	111 ± 1
3F_5	76	83.8	3.1 ± 0.1	6.8 ± 0.1	1.9	2.2 ± 0.1	108 ± 1
3F_6	83	90.0	4.1 ± 0.3	12.5 ± 4.7	55.4	3.3 ± 0.1	102 ± 1
3F_7	80	66.4	5.7 ± 0.3	433.5 ± 303.3	76.0	2.5 ± 0.1	101 ± 1
3F_8	79	85.8	5.2 ± 0.1	10.9 ± 1.3	1.8	2.8 ± 0.3	100 ± 2
3F_9	87	80.6	5.1 ± 0.1	292.0 ± 156.6	57.6	2.1 ± 0.1	111 ± 1
3F_10	79	85.6	4.7 ± 0.1	9.2 ± 0.1	1.6	2.2 ± 0.2	113 ± 1
3F_11	63	81.3	5.9 ± 0.9	403.9 ± 307.1	68.7	2.9 ± 0.8	109 ± 2
3F_12	69	82.2	4.0 ± 0.1	7.8 ± 0.1	1.6	3.0 ± 0.1	117 ± 1
2F_1	74	83.5	2.6 ± 0.1	5.1 ± 0.1	1.6	3.2 ± 0.5	109 ± 1
2F_2	74	78.5	2.5 ± 0.1	5.1 ± 0.1	1.6	3.6 ± 0.4	100 ± 1
2F_3	75	57.0	3.0 ± 0.1	5.9 ± 0.1	1.7	4.7 ± 0.1	93 ± 1

T_out_: Spray dryer outlet temperature, T_g_: Glass transition temperature.

**Table 3 pharmaceutics-12-01116-t003:** Theophylline aerosolization performances of the spray-dried formulations.

Sample ID	TH FPF < 5 µm (%)	TH FPF < 3 µm (%)	MMAD (µm)	GSD
3F_3	43.1 ± 0.1	24.2 ± 0.1	7.70 ± 0.1	3.6 ± 0.1
3F_6	34.1 ± 0.2	17.6 ± 0.2	13.1 ± 0.2	5.5 ± 0.1
3F_9	18.4 ± 0.1	7.2 ± 0.1	60.2 ± 0.1	12.3 ± 0.1
3F_12	26.4 ± 0.1	12.1 ± 0.1	24.6 ± 0.2	8. 3± 0.1
2F_3	54.3 ± 0.3	39.5 ± 0.2	3.70 ± 0.1	2.2 ± 0.1

FPF: Fine particle fraction; MMAD: Median mass aerodynamic diameter; GSD: Geometric standard deviation.

**Table 4 pharmaceutics-12-01116-t004:** Salbutamol sulphate aerosolisation performances of the spray-dried formulations.

Sample ID	SS FPF < 5 µm (%)	SS FPF < 3 µm (%)	MMAD (µm)	GSD
3F_3	36.0 ± 0.3	17.7 ± 0.3	13.3 ± 0.7	5.1 ± 0.1
3F_6	24.4 ± 2.3	10.5 ± 0.2	31.3 ± 2.3	8.6 ± 0.4
3F_9	15.6 ± 0.9	5.2 ± 0.2	229.5 ± 38.6	12.3 ± 0.1
3F_12	24.7 ± 0.1	10.8 ± 0.1	38.0 ± 1.6	11.6 ± 0.2
2F_3	54.9 ± 0.3	39.2 ± 0.1	3.8 ± 0.1	2.3 ± 0.1

FPF: Fine particle fraction; MMAD: Median mass aerodynamic diameter; GSD: Geometric standard deviation.

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
