# Peer review of "The Use of a Three-Fluid Atomising Nozzle in the Production of Spray-Dried Theophylline/Salbutamol Sulphate Powders Intended for Pulmonary Delivery"

_pharmaceutics, 2020, doi:10.3390/pharmaceutics12111116_

Round 1
Reviewer 1 Report
Dear Authors,
Please see my comments below:
1. Introduction:
a. suggest to start a new paragraph from Line 39
b. suggest to combine Paragraph Line 59-61 and its previous paragraph
c. please reorganize paragraphs and use proper connecting or transition words
d. you introduced backgrounds of TH but did not mention the other API in your formulation, SS. Please consider adding (a) paragraph(s) to introduce SS. Also, please introduce the clinical significance of combining TH and SS
2. Method
Line 218: should be um
3. Result
a. Line 291-292: which spray drier was used, 2F or 3F? Please clarify
b. Please add footnote to Table 2 to explain the acronyms
c. Figure 3: there are peak shift in 3F 8 and 9. Please double check and explain if the shifts are real
d. Line 349-350: typically, size range of 1-5 um is considered suitable for pulmonary delivery. Please consider using '... indicates most of them are suitable for pulmonary drug delivery'
c. how is the quantitative homogeneity of TH and SS in each particles? And how is the distribution of TH and SS in particles (e.g. location of these two drugs inside each particle)
d. Please present the exact values of aerodynamic parameters for each run (e.g. MMAD, GSD, FPF, ED, etc.)
4. Other
What is the advantage of using 3F SD over physically mixing inhalable powders produced by 2F SD? It would be ideal if the core-shell structure is successfully achieved.
Author Response
- Introduction:
- suggest to start a new paragraph from Line 39
We agree and have updated the text accordingly.
- suggest to combine Paragraph Line 59-61 and its previous paragraph
Thank you for your comment. The paragraphs have been combined in the revised manuscript.
- please reorganize paragraphs and use proper connecting or transition words.
Thank you for your comment. have been made clearer and reorganised carefully.
- you introduced backgrounds of TH but did not mention the other API in your formulation, SS. Please consider adding (a) paragraph(s) to introduce SS. Also, please introduce the clinical significance of combining TH and SS.
Thank you for your suggestion. The following has been added to the text of the revised manuscript: “Salbutamol sulphate was chosen as it is a widely used short-acting β2-agonist in the symptomatic relief of asthma and COPD and is frequently prescribed as maintenance therapy. A combination of inhaled SS and TH could provide greater efficacy for patients who remain symptomatic on a monotherapy with SS. The effectiveness of this combination therapy has been already established in the literature; for example, Nishimura et al. demonstrated a superior bronchodilation effect by combining inhaled SS and orally delivered TH compared to the use of the individual agents alone.”
- Method
Line 218: should be um
We apologise for this mistake. The error has been corrected.
- Result
- Line 291-292: which spray drier was used, 2F or 3F? Please clarify
Both the pure drugs were spray dried using the 2F nozzle. The sentence has been modified as follows: “SS spray dried alone from water, using the 2F nozzle, resulted in spherical particles with dimpled surfaces….” to make this point clearer.
- Please add footnote to Table 2 to explain the acronyms
A footnote to explain the meaning of Tout and Tg has been added.
- Figure 3: there are peak shift in 3F 8 and 9. Please double check and explain if the shifts are real
This observation is correct. We are sorry for the mistake. We had plotted the XRPD diffractograms that were obtained by two different run methods. The previous incorrect figure has now been replaced with the correct one.
- Line 349-350: typically, size range of 1-5 um is considered suitable for pulmonary delivery. Please consider using '... indicates most of them are suitable for pulmonary drug delivery'
Thank you for your suggestion. The sentence has been modified accordingly.
- how is the quantitative homogeneity of TH and SS in each particles? And how is the distribution of TH and SS in particles (e.g. location of these two drugs inside each particle)
Thank you for the question.
In this work we assumed that the particle composition reflected the theoretical values of their weight ratio as introduced to the spray dryer, namely 7:3. 8:2 and 9:1. While HPLC analysis was not performed on the samples prior to powder characterisation and deposition studies, nevertheless PXRD and IR analyses showed a linear decrease/increase of the peaks associated with the two drugs relating to the amount of the APIs in the powders, providing indirect evidence of the sample composition. Nevertheless, we recognize this limitation should be mentioned in the paper, so we added the following sentence at the end of the ‘results and discussion’ section: “Nevertheless those results were obtained assuming that the particle compositions reflected the theoretical values based on the calculated weight ratio, namely 7:3. In the light of this, further investigations and powder optimisation are required.”
Regarding the distribution of TH and SS particles: the only analysis we performed to understand the real position of the two drugs was by visual inspection via SEM. We appreciate that this technique is not sufficient to investigate the molecular distribution of the components within the particles. For this purpose a possible approach would be to conduct EDX analysis, however this approach is generally not recommended for particles of this size range since the area of analysis and the X ray penetration depth are generally much higher than the actual particle dimension. We agree with the reviewer that further elaboration on this point would be helpful and further analysis will be performed in the future.
- Please present the exact values of aerodynamic parameters for each run (e.g. MMAD, GSD, FPF, ED, etc.)
Thank you for your suggestion. Two tables (which were inadvertently omitted from the original manuscript) have now been added to describe the aerodynamic parameters of SS and TH.
- Other
What is the advantage of using 3F SD over physically mixing inhalable powders produced by 2F SD? It would be ideal if the core-shell structure is successfully achieved.
We thank the Reviewer for the question.
In general, a 3F nozzle is useful to produce binary systems in which two drugs have different solubilities. Nevertheless, the approach described in the manuscript, namely the variation of the relative flow speed between the solutions pumped in the inner and outer nozzle channels, allowed co-formulations to be obtained with a wide range of possibilities in terms of drug ratio as well as fine tuning of the drug component in the final product. In addition, this concept is particularly advantageous when compounds with a low solubility in both hydrophobic and hydrophilic solvents, such as theophylline, are being formulated. In fact, this strategy was useful to increase the concentration of theophylline within the powders, overcoming the issues related to its solubility limits.
As the Reviewer correctly affirmed, the ideal situation would be to obtain a core-shell micro-structure in which the outer theophylline crystals protect the inner salbutamol sulphate core from the environmental moisture in the same way as the hydrophobic aminoacids protect the components of a powder intended for pulmonary delivery (Mah et al. Eur. J. Pharm. Biopharm., 144 (2019) 139-153). This was the first idea behind the work conceptualisation as stated in the text. However, preliminary SEM analysis revealed that powders appeared to be composed mainly of a mixed binary system comprised of theophylline nano-crystals and separate salbutamol sulphate dimpled particles. We believe that a core-shell structure could be achieved by adding one or more ingredients in the theophylline-containing solution prior to spray drying but this strategy is contrary to one of the aims of this work which is to produce excipient free formulations. In any case, a new study is already in progress in order to understand what are the main factors that affect the particle structures using a 3F nozzle.
Reviewer 2 Report
The manuscript albeit potentially interesting seem to address only partially the problem of preparation of DPI suitable theophylline/salbutamol excipient-free co-spray-dried powders. As it is, the paper is more suitable as a communication rather than a full research article. Additional investigation is required in order to wholly address this issue.
Here below are the main perplexities.
1) The main problem is that the experimental design seems insufficient to unravel the problem and it is unclear how most conditions were chosen. perhaps a design of experiment approach would help clearing this issue.
2) It is not clear how the AA have chosen the preparation conditions. Why did the AA not compare 2-F and 3-F formulations in the same conditions? 2-F were performed only with water. Naturally, drug solubilities in the different solvents determine the choice. However, both drugs are at least sparingly soluble in the solvents chosen. This may imply rethinking of the experimental design.
3) In light of the scarce aerodynamic performances, why were a small amounts of excipients (lubricants or porigens) tested to improve the aerodynamic behavior? This would not undermine the value of the present approach.
2) Fig 4 is not very informative, Fig 5 may be enough to show the Tg transition.
3) Fig 6 is not clear and easily readable. Better if may signals are reported in a table to show main differences and commented in the text.
Author Response
The manuscript albeit potentially interesting seem to address only partially the problem of preparation of DPI suitable theophylline/salbutamol excipient-free co-spray-dried powders. As it is, the paper is more suitable as a communication rather than a full research article. Additional investigation is required in order to wholly address this issue.
Here below are the main perplexities.
1) The main problem is that the experimental design seems insufficient to unravel the problem and it is unclear how most conditions were chosen. perhaps a design of experiment approach would help clearing this issue.
We thank the Reviewer for their feedback. The conditions were chosen based on in which the spray drying parameters chosen allowed particles in the respirable size range to be obtained. The parameters investigated were: spray drier inlet temperature, spray drier aspiration pump rate and feed flow rates. This is indicated in lines 124-126 of the revised manuscript.
As stated in the conclusion section (lines 479-481) further investigations are required to optimise the method and a DOE approach could be helpful in this sense. The current work was mainly focused on the effect of the solvent used and the flow rate ratio between the two phases, as well as on the concept that the particles obtained by a 3F nozzle could be potentially useful for lung delivery, overcoming issues related to low compound solubility and, in turn, the dosage limitations. In light of that, we believe that the points mentioned above have been addressed by the study. However, new studies are already in progress in order to address the open questions. Again, this is indicated in the text of the revised manuscript (lines 124-126).
2) It is not clear how the AA have chosen the preparation conditions. Why did the AA not compare 2-F and 3-F formulations in the same conditions? 2-F were performed only with water. Naturally, drug solubilities in the different solvents determine the choice. However, both drugs are at least sparingly soluble in the solvents chosen. This may imply rethinking of the experimental design.
We thank the Reviewer for the question.
SS does not have a water solubility limitation using the experimental conditions described in the text hence we preferred to restrict the use of organic solvents to the TH. Water is generally preferred in both small scale and industrial scale for the production of respirable particles since solvent traces above a certain level could result in toxicity to the lungs. For this reason, only water was used in the 2F nozzle experiment given that other solvents did not significantly increase the amount of TH that could be solubilised. Studies using the 3F nozzle but with water in channels both channels (for TH and SS) were also undertaken for direct comparison (Table 1), such that the effect on the final product when the compounds were spray dried from the same solvent could be studied.
Following the same concept, water miscible (ethanol and acetone) and immiscible solvents (chloroform) were also used to further investigate the impact of solvent miscibility on the product. This procedure allowed us to study different conditions regarding the solvent effect.
Nevertheless, we recognize this concept should be mentioned in the paper, so we added the following sentence to the revised manuscript (line 281): “It is worth noting that the solvents chosen in this study allowed us to monitor the effect of liquids having different affinities with water, since, as well as using water alone, both water miscible (ethanol and acetone) and water immiscible (chloroform) solvents were used for the TH feed solution.
3) In light of the scarce aerodynamic performances, why were a small amounts of excipients (lubricants or porigens) tested to improve the aerodynamic behavior? This would not undermine the value of the present approach.
We appreciate the Reviewer’s insightful suggestion. As stated in the first comment, the focus of the work was primarily on the novelty of the use of the 3-fluid nozzle atomiser and on assessment of the approach described. Nevertheless, we have already scheduled new studies based on different approaches in order to improve the final product characteristics, especially its aerodynamic behaviour. These studies will align with our previous work where we have used amino acids to improve powder performance (Mah et al. Eur. J. Pharm. Biopharm., 144 (2019) 139-153 and Focaroli et al. Int. J. Pharm. 562 (2019) 228–240).
4) Fig 4 is not very informative, Fig 5 may be enough to show the Tg transition.
We thank the reviewer for the suggestion. The figure has been deleted from the main manuscript and it has been added to the supplementary materials section.
3). Better if may signals are reported in a table to show main differences and commented in the text.
We thank the Reviewer for pointing this out.
We have increased the resolution of figure 6.
Although we agree, in general, that an in-depth FTIR analysis would be useful in in order to highlight the presence of new functional groups and/or chemical entities, we believe that the information we have provided is sufficient in this particular case for the characterisation of our formulations since no extra peaks were found. In this work the only information that can be obtained by the FTIR analysis is related to the production of amorphous material, as already described in the text. For this reason, we respectfully disagree with the Reviewer since we believe that the addition of a signal list does not give any additional useful information relating to the formulation characteristics.
Reviewer 3 Report
Dear Authors,
This is interesting work. A few comments below
Section 2.2. Why was TH only used at 1 concentration and again the parameters were fixed for spray drying. Could TH at different concentrations and spraydrying parameters been investigated to optimise formulation and aerosolization
Section 2.9. check units of aerodynamic diameters line 217-218 – should be micrometres. Also line 214 the HPLC method is section 2.10. Could the filters affect airflow and hence deposition? Why not just coat plates directly? How many capsules where aerosolised per replicate?
Section 2.10 was limit/minimum detection of SS and TH determined
Is table 2 discussed within text? How was residual solvent content determined?
Results and discussion will benefit from subheadings
Where is table 3 & 4 for aerosolization
Author Response
Section 2.2. Why was TH only used at 1 concentration and again the parameters were fixed for spray drying. Could TH at different concentrations and spra ydrying parameters been investigated to optimise formulation and aerosolization
We thank the Reviewer for the question. The spray drying parameters have been chosen based on preliminary experiments in which the spray drying parameters chosen allowed particles in the respirable size range to be obtained. In this work we mainly focused on the effect of the solvent used and the flow rate ratio between the two phases. However, a study to determine the role of other parameters on the characteristics of the final product has been already scheduled.
We used only one TH concentration since it represented the highest amount of the drug that could be solubilised in the different solvents in a reasonable timeframe. A lower TH concentration would have led to a higher solvent consumption and consequently to a higher products waste.
Section 2.9. check units of aerodynamic diameters line 217-218 – should be micrometres. Also line 214 the HPLC method is section 2.10. Could the filters affect airflow and hence deposition? Why not just coat plates directly? How many capsules where aerosolised per replicate?
We apologise for the mistakes; the text has been changed accordingly.
Regarding the filters: this method has been already used in the past by our research group (Pei T. Mah et al. E J Pharm, 144, 2019, p:139-153). The filters were used to minimise the “bouncing effect” and it has been proven to be more reliable than the direct plates coating in terms of result reproducibility.
We used 3 capsules containing 10 mg of powder for each replicate. This is indicated in the manuscript in line 225
Section 2.10 was limit/minimum detection of SS and TH determined
We thank the Reviewer for the question. The SS limit of detection was 12 mcg/ml whereas the TH limit wars 5 mcg/ml. The values have been added in section 2.10 (line 251).
Is table 2 discussed within text? How was residual solvent content determined?
Table 2 is mentioned in line 311. The residual solvent content was determined by Thermogravimetric analysis (TGA, section 2.8).
Results and discussion will benefit from subheadings
We appreciate the reviewer’s insightful suggestion; this section has now been divided into subchapters/subheadings.
Where is table 3 & 4 for aerosolization.
We apologise for our error in inadvertently omitting these tables. The tables have been added to the revised manuscript
Round 2
Reviewer 2 Report
The manuscript seems clearer now and could be accepted after minor revision.
The reason is that I'm still convinced that FTIR data do not add important information as they cannot show insights into possible interactions due to signal broadening and therefore they do not provide any additional information compared to those already provided by thermal analysis and XRPD.
As a prove, the discussion of FTIR data is rather limited to a few lines that could be actually omitted without any change in the manuscript value.
I suggest to move FTIR analysis to supplementary materials.
The language requires some check for grammar errors/typos.
Author Response
I suggest to move FTIR analysis to supplementary materials.
We thank the Reviewer for the suggestion. We added the FTIR analysis to the “supplementary materials” section and the text has been changed accordingly.
The language requires some check for grammar errors/typos.
Thank you for the advice. We have carefully reviewed the manuscript and made some further grammatical and typographical corrections, as indicated in the revision submitted.